# Malnutrition Prevention after Allogeneic Hematopoietic Stem Cell Transplantation (alloHSCT): A Prospective Explorative Interventional Study with an Oral Polymeric Formulation Enriched with Transforming Growth Factor Beta 2 (TGF-β2)

**DOI:** 10.3390/nu14173589

**Published:** 2022-08-31

**Authors:** Enrico Morello, Francesco Arena, Michele Malagola, Mirko Farina, Nicola Polverelli, Elsa Cavagna, Federica Colnaghi, Lorenzo Donna, Tatiana Zollner, Eugenia Accorsi Buttini, Marco Andreoli, Chiara Ricci, Alessandro Leoni, Emanuela Samarani, Alice Bertulli, Daria Leali, Simona Bernardi, Domenico Russo

**Affiliations:** 1Department of Clinical and Experimental Sciences, University of Brescia, Bone Marrow Transplant Unit, ASST-Spedali Civili di Brescia, 25123 Brescia, Italy; 2Dietetics and Clinical Nutrition Unit, ASST-Spedali Civili Brescia, 25123 Brescia, Italy; 3Gastroenterology Unit, ASST-Spedali Civili Brescia—University of Brescia, 25123 Brescia, Italy; 4Central Laboratory, ASST-Spedali Civili Brescia, 25123 Brescia, Italy

**Keywords:** malnutrition, PG-SGA, GVHD, allogeneic hematopoietic stem cell transplantation, TGF-β2

## Abstract

Malnutrition is common after allogeneic Hematopoietic Stem Cell Transplantation (alloHSCT), and interventions directed to correct nutritional status are warranted to improve transplant outcomes. In this prospective study, an oral polymeric formulation enriched with TGF-β2 (TE-OPF) was explored to correct malnutrition according to Patient-Generated Subjective Global Assessment (PG-SGA). TE-OPF was proposed to 51 consecutive patients who received transplants at our institution for hematological malignancies, and sufficient dose intake was established per protocol as at least 50% of the prescribed dose of TE-OPF: group A received adequate nutritional support; group B, inadequate. The study met the primary outcomes in terms of safety (no adverse events reported during TE-OPF intake except for its disgusting taste) and malnutrition (PG-SGA C 28 days after transplant): severely malnourished patients (PG-SGA C) accounted for 13% in group A and 88.9% in group B (*p* = 0.000). At the end of the study, after a median follow-up of 416 days, the estimated median Overall Survival (OS) was 734 days for well or moderately nourished patients (PG-SGA A/B) in comparison to 424 for malnourished patients (*p* = 0.03). Inadequate TE-OPF intake was associated with an increase in acute gastrointestinal Graft Versus Host Disease (GVHD) cumulative incidence (38% vs. 0% *p* = 0.006). A higher incidence of pneumonia was reported in group B (*p* = 0.006). IGF-1 levels at 14 and 28 days after transplant were significantly higher in group A and were associated with a lower incidence of acute GVHD (aGVHD). Higher subsets of B, T, and NK cells were found in group A, and a higher number of CD16+ NK cells was associated with a lower incidence of acute GVHD (*p* = 0.005) and increased survival at the end of the study (*p* = 0.023). Artificial neural network analysis suggested that inadequate TE-OPF intake, pneumonia, and sepsis significantly affected malnutrition 28 days after alloHSCT and survival 365 days after alloHSCT (normalized importance 100%, 82%, and 68%, respectively). In this exploratory and preliminary study, the use of TE-OPF appeared to reduce the incidence of malnutrition after alloHSCT, but larger and controlled studies are required.

## 1. Introduction

The symptomatic burden of allogeneic hematopoietic stem cell transplantation (allo-HSCT) can severely affect food intake and nutritional status [1,2,3,4]. Severe malnutrition puts transplant outcomes at risk, prolonging the time to engraftment and increasing the risk of infection, as well as the duration of hospitalization and mortality [5,6,7,8].

Nutritional assessment and nutritional support are therefore crucial to manage transplant patients and improve the transplant outcome [7,8,9,10]. This aspect is even more important if we consider that over the last ten years, the age limit for transplantation has risen to the age of 75, and patients older than 60 years old who have received transplants exceed 30% [11].

The Patient-Generated Subjective Global Assessment (PG-SGA) questionnaire is recommended as a standard tool for nutritional screening, assessment, monitoring, and triaging for nutritional interventions in patients with cancer [12].

Regarding nutritional support, enteral nutrition (EN) is generally considered the best approach in oncology, but, if that cannot be pursued, parenteral nutrition (PN) is recommended [13]. In HSCT, PN is generally reserved in the case of severe mucositis (Grade 3–4), ileus, or intractable emesis [13] and should be continued until the resolution of the complications [14]. Although ASPEN and ESPEN guidelines recommend EN as the best approach in nutritional support in HSCT [14,15], parenteral nutrition is still most widely used in HSCT settings [16]. Furthermore, it is well known that PN may be associated with gut mucosal atrophy, metabolic and hepatic complications, central venous catheter infections, fluid overload, and hyperglycemia, which also increase the risk of systemic infections and inflammation [10,16,17,18,19].

GVHD is the main cause of transplant-related morbidity and mortality after disease relapse [20]. From a clinical and pathogenetic point of view, gastrointestinal (GI) acute GVHD has many aspects in common with chronic inflammatory bowel disease (IBD), such as the loss of the intestinal epithelial barrier, alterations in the intestinal microbiota, and the use of immunosuppressive therapies for clinical interventions [20,21,22,23,24,25,26,27]. In an IBD setting, TGF-β2 is the most studied bioactive peptide for nutritional support [28]. TGF-β2 is an anti-inflammatory cytokine and a key modulator of the microbiota. It also has a relevant role in host immune cell crosstalk, it controls the differentiation, proliferation, and activation state of lymphocytes, macrophages, and dendritic cells, and it plays a critical role in the mechanisms of tolerance, prevention of autoimmunity, and in anti-inflammatory processes in the gut [28,29].

In this prospective interventional study, 51 patients who consecutively underwent allo-HSCT were evaluated from a nutritional point of view and supported with an Oral Polymeric Formulation Enriched with TGF-β2 (TE-OPF), commonly used in IBD settings.

## 2. Materials and Methods

The study was approved by the local ethical committee in March 2020. Patients signed informed consent and underwent nutritional evaluation by a dietitian. PG-SGA score nutritional support with an oral polymeric formulation enriched with TGF-β2 (MODULEN-IBD^®^) was proposed to all enrolled patients. Patients older than 18 years submitted to alloHSCT were included in the study, while exclusion criteria were due to unsigned informed consent, gastrointestinal perforations and fistulas, and intractable vomiting. The primary outcome of the study was to reduce the incidence of severe malnutrition (PG-SGA C, see below) 28 days after allo-HSCT to less than 50% in comparison to a historical control (in which the incidence of severe malnutrition at +28 days after allo-HSCT was 75% [30]. The study was closed when at least 24 subjects had taken at least 50% of the prescribed TE-OPF (sample size calculation with a power of 75% and an alpha level of 0.05). The baseline characteristics of the 51 enrolled patients are displayed in Table 1.

Malnutrition impact and clinical outcomes were explored according to TE-OPF intake in percentage based on the prescribed dose (treatment ratio—TR). No minimal dose was defined as clinically effective, but in the protocol design, 50% of the prescribed TE-OPF was considered adequate (Group A, Treatment Ratio TR > 50% of the prescribed dose); Group B was comprised of patients who did not take an adequate dose of TE-OPF. The Mann–Whitney test was used for statistical analysis regarding the correlations among TR and malnutrition, acute GVHD onset, clinical events (sepsis or pneumonia), and biomarker studies; for biomarker studies, Kaplan–Meier plots were defined according to the 25th percentile rounded to the nearest whole number. Overall survival 365 days after transplant was estimated according to Kaplan–Meier curves, and the log-rank test was used for univariate analysis. The cumulative incidence of acute GVHD was calculated months months after transplantation. In the log-rank and Fisher tests, patients were grouped according to the treatment ratio (TR) (TR> or <50% based on the study protocol, Group A and Group B, respectively). No differences were found in patients’ characteristics at admission for transplant between the groups (Table 2).

Nutritional status was assessed through the PG-SGA questionnaire [12,14] at admission and on day 0, +7, +14, +21, and +28 from transplant (day 0 was the transplant day). PG-SGA is composed of an objective section and a patient-reported one. The first reports the overall nutritional status with an alphabetical score (A = good nutritional status; B = moderate malnutrition; C = severe malnutrition). The second one is a numeric score that is calculated from four items reported by patients (weight loss, food intake, symptoms with a nutritional impact, and physical activity). Integration with Parenteral Nutrition (PN) was permitted when oral food intake was less than 60% of hospital meals for at least three days and/or oral mucositis grade 3–4, intractable diarrhea, and emesis were present, according to the criteria reported in ESPEN Guidelines 2009 [13].

Full blood counts, flow cytometric lymphocyte subpopulations’ absolute count (28 days after transplant), and serum IGF-1 levels with Diasorin “Liaison IGF-1” chemoimmunoluminescence analysis (14 and 28 days after transplant) were explored as potential biomarkers according to preliminary data [31]. Continuous variables were categorized at the 25th percentile for log-rank and Chi-square test analyses.

Multivariable analysis was performed with artificial neural network analysis in order to explore the relative importance of the several variables for clinical outcomes (malnutrition and survival at 365 days after transplantation [32].

### TE-OPF Treatment Plan

The TE-OPF was diluted in a bottle of mineral water and proposed one or more times per day if necessary. The nutritional support started at admission and was carried out until discharge from allo-HSCT. Refusal and reasons for refusal of the proposed treatment were recorded. The amount of calories derived by oral supplement was calculated at admission based on on BMI and total daily energy expenditure (TDEE) by multiplying the basal metabolic rate (the Mifflin St. Jeor BMR estimation formula was used) by 1.3 (Ppysical activity level). From admission to the day of transplant, the amount of calories derived from oral formulation enriched with Transforming Growth Factor beta 2 was as follows:-BMI less than 22: 20% of TDEE;-BMI between 22.1 and 24.9: 12% of TDEE;-BMI between 25 and 29.9: 10% of TDEE;-BMI between 30 and 34.9: 8% of TDEE;BMI higher than 35: 5% of TDEE.

From day 1 to day +28 from transplantation, the quantity of supplement was increased by 10% compared to the initial dose if the patient showed a weight loss of less than 5% during hospitalization; with a weight loss of more than 5%, the dose of oral formulation was increased by 20% compared to the starting dose. Once reconstituted by the staff, the supplement could be flavored according to the patient’s taste with barley coffee or decaffeinated coffee. Compliance with TE-OPF intake was investigated with a specific questionnaire, and the percentage of administered dose compared to the prescribed dose was registered daily (Treatment Ratio—TR). Adverse events were reported according to good clinical practice (GCP).

## 3. Results

Primary declared outcomes were safety of TE-OPF and malnutrition 28 days after transplantation defined as PG-SGA C 28 days after transplantation.

No adverse events were attributed to TE-OPF. Patients refused the preparation due to a disgusting sensation in 19/27 cases (70%), mucositis in 10/27 (37%) or respiratory complications involving the use of non-invasive ventilation in 5/27 (18.5%). The mean assumption according to the prescribed dose (at least 50%), was 60%, median 46%, minimum 3% to maximum 224%.

The percentage of severely malnourished patients (PG-SGA C) after 28 days was significantly lower in the group with a treatment ratio (TR) higher than 50% (GROUP A) in comparison to the group with insufficient TE-OPF intake (GROUP B—TR < 50%) (12.5% vs. 88.9% *p* = 0.000) (Table 1). The study met the outcome criteria and was stopped.

In group A, 7 out 20 patients (30.4%) were well nourished (PG-SGA A) in comparison to 0 in the group with an inadequate intake of TE-OPF (Group B). No statistical differences were found in baseline patients’ characteristics at admission in both groups (Table 2). No statistically significant correlation was found between the type of conditioning and nutritional status at day +28 from allo-HSCT.

The percentage of the prescribed dose ranged from 3% to 224%, with a median of 46%. Higher TR, less subjective PG-SGA values (patient-reported numeric score) were registered 28 days after alloHSCT (correlation R2 0.288, *p* = 0.000) (Figure 1).

The mean percentage of TR (TE-OPF assumption as a percentage of the prescribed dose) in the group with PG-SGA scores A, B, and C at 28 days after transplantation was 129%, 85%, and 27% respectively. The difference was proved to be statistically significant (between A and B, *p* = 0.045 and between both A + B and C, *p* = 0.000; Mann–Whitney test) (Figure 2).

Study analysis was performed one year after the last patient enrolment with a median follow-up of 416 days (29–784). The estimated median OS of the whole population was 649 days (456–841), whereas patients in group A experienced an estimated median survival of 734 days in comparison to 424 in group B (*p* = NS). The estimated median OS was 734 days for patients achieving the primary study outcome (well or moderately nourished 28 days after transplant—PG-SGA A/B) in comparison to 424 for malnourished patients (*p* = 0.03) (Figure 3).

### 3.1. Secondary Outcomes

#### 3.1.1. TE-OPF Assumption and GVHD

The estimated cumulative incidence of aGVHD Magic B or higher according to the MAGIC Consortium Criteria was similar in patients treated with adequate TE-OPF (Group A) in comparison to patients who received less than 50% of the prescribed dose (19% vs. 44% *p* = NS). Gastrointestinal GVHD (all grade) was reported only in patients belonging to group B (cumulative incidence of 38% vs. 0% *p* = 0.006) (Figure 4). Mean TE-OPF assumption was 68.3% of the prescribed dose in patients who did not experience acute GVHD and 17.8% in patients with aGVHD (Figure 5, *p* = 0.002).

Chronic GVHD cumulative incidence was 20%, and no differences were reported in the different treatment groups.

#### 3.1.2. TE-OPF Assumption and Infectious Complications

Pneumonia was more frequent in group B patients (*p* = 0.006, 48.1% vs. 12.5%). Sepsis and enteritis were more frequent in group B, but the difference was not significant (55.5% vs. 29.1%, 37% vs. 16.6%, respectively) (Table 3).

#### 3.1.3. TE-OPF Assumption and Relapse-Free Survival

The relapse incidence was 25.9% in group A and 29.1% in group B (*p* = NS).

### 3.2. Biomarker Studies

#### 3.2.1. TE-OPF and IGF-1

The mean IGF-1 was higher in group B at 14 and 28 days after transplantation (109 vs. 165 and 109 vs. 153, *p* = 0.024 and 0.013, respectively; normal range 99–364 ng/mL). Cumulative incidence of aGVHD was lower if values of IGF-1 at 14 days after transplant were higher than 100 ng/mL (*p* = 0.045). Cumulative incidence of acute gastrointestinal GVHD was lower in patients with higher values of IGF-1 at 14 and 28 days (*p* = 0.021 and 0.032, respectively).

#### 3.2.2. TE-OPF, Malnutrition, and Lymphocyte Subsets

In group A, levels of total lymphocytes, B, T, and NK cells (mature and immature) 28 days after alloHSCT were higher than those in group B as reported in Table 4. The difference was significant (*T*-test) for CD16+ NK cells (*p* = 0.03) and CD3+ /CD4+ T cells (*p* = 0.025). Well-nourished patients (PG-SGA A and B) experienced a significantly higher number of CD16+ NK cells (*p* = 0.016), and a higher number of CD16+ NK cells (>100/mcL) was associated with a reduction in the cumulative incidence of acute GVHD (20% vs. 69% at 120 days after transplantation, *p* = 0.005) and an increased survival (median not reached vs. 734 days, *p* = 0.023).

Multivariable analysis was performed according to the method described by Caocci et al. [32] through artificial neural network analysis, and the normalized importance of several variables is reported: group A, pneumonia and sepsis significantly affected malnutrition 28 days and survival after one year (normalized importance: 100%, 82%, and 68%, respectively).

## 4. Discussion

The main objective of this prospective interventional study was to evaluate the incidence of severe malnutrition (PG-SGA C) at 28 days after alloHSCT in a cohort of 51 patients supported, from admission to day +28, with an oral polymeric nutritional supplement enriched with TGF-β2.

The use of this support is approved by the ACBS and NICE for “Crohn’s Disease active phase, and in remission if malnourished”, and this is the first prospective study to test it in the setting of malnutrition after alloHSCT.

One randomized clinical study showed that exclusive enteral nutrition with MODULEN-IBD^®^ is more efficient to achieve mucosal healing than corticosteroids, proving in addition that this TE-OPF has trophic, anti-inflammatory, and microbiota-modulating effects on the gastrointestinal tract [33]. The anti-inflammatory effect of MODULEN-IBD^®^ is attributable to the high levels of TGF-β: an immunosuppressive cytokine involved in the development and functions of immune cells, including T and B cells and also dendritic cells (DCs) [29].

In this prospective study, early assumption of TE-OPF was feasible and safe. Patients who took at least the half of the prescribed protocol dose achieve a reduced incidence of severe malnutrition 28 days after transplant according to the study design. The mean percentage of the Treatment Ratio (TR) in the group with PG-SGA score A at +28 days from allo-HSCT (well nourished) was 129%, in the group with PG-SGA score B (moderate malnutrition), was 85%; and in the group with PG-SGA C (severe malnutrition), 27%. The difference was proved to be statistically significant (between PG-SGA A and PG-SGA B, *p* = 0.045 and between both PGSGA A + B and PG-SGA C, *p* = 0.000; Mann–Whitney Test) and the effect on nutritional status may be dose dependent (Figure 1). In the subjective load of gastrointestinal symptoms, the higher the intake of TE-OPF, the lower the PG-SGA numeric score (Figure 2); the higher the score, the worse the nutritional status of the patients.

Acute GVHD was reported more frequently in patients with an inadequate assumption of TE-OPF, especially gastrointestinal GVHD that was absent in patients who took at least 50% of the prescribed dose. These preliminary results seem to confirm that the effect of this TE-OPF could be similar to that observed in IBD due to microbiota-modulating, anti-inflammatory, and trophic effects. As explained in the introduction, in both IBD and alloHSCT settings, malnutrition and the immunity response of the gastrointestinal tract are relevant. From both a clinical and pathogenetic point of view, gastrointestinal acute GVHD shares many aspects with chronic inflammatory bowel diseases (IBD), such as the loss of the intestinal epithelial barrier; alterations in the intestinal microbiota; cascade of T cell activation, proliferation, and cytotoxic activity; and the use of immunosuppressive therapies [22,25]. In bone marrow transplantation, acute GVHD may appear in 20–40% of patients with matched related donors, and in more than 50% of haploidentical stem cell transplants. Regarding gastrointestinal acute GVHD, up to 50% of patients with acute GVHD experience gastrointestinal symptoms [34]. All of these observations allow us to hypothesize that MODULEN-IBD^®^ may also reduce the risk of GI GVHD thanks to its trophic, anti-inflammatory, and microbiota-modulating effects on the gastrointestinal tract. Reducing GI GVHD cumulative incidence was not the primary goal of the study, but patients who took more than 50% of the prescribed TE-OPF did not experience GI aGVHD (Figure 4), and the effect seemed to be dose dependent (Figure 5). Explaining the reasons with these data is not easy. It may be essential to evaluate the role of a prophylactic and earlier oral intake of MODULEN-IBD^®^ from admission to +28 days from admission. This choice probably avoided or reduced microbiota dysbiosis: an important risk factor for GI GVHD [35,36,37]. Furthermore, the loss of microbiota diversity and the reduction of short chain fatty acid production have been observed in patients with GVHD [35,36,37]. Acetate, propionate, and butyrate are metabolites produced by microbiota and play an important role in the interaction between the microbiota and host immune cells, influencing systemic autoimmune responses and participating in different steps of inflammation processes [35,38,39]. MODULEN-IBD^®^ administration in patients submitted to allo-HSCT may reduce the risk of GI GVHD due to its microbiota-modulating, anti-inflammatory, and trophic effects. Regarding intestinal trophism, patients with mucositis often refuse food and this further worsens intestinal dysbiosis. However, thanks to the consistency of MODULEN-IBD^®^, patients often continue to take an oral nutritional supplement despite oral pain, avoiding fasting. This may allow patients to maintain microbiota diversity.

Biomarkers’ data seemed to confirm the role of IGF-1 [31] as potential biomarker in this setting, suggesting that higher values (more frequent in well-nourished patients) seem to be protective for acute gastrointestinal GVHD, as reported for other chronic illness [40,41,42,43]. An increase in lymphocytes was reported according to TE-OPF consumption, which was more evident for NK subpopulations. Expansion of mature CD16+ NK in this population seemed to correlate with clinical outcomes such as acute GVHD and survival one year after transplant: recently, a model of NK expansion after UCB transplantation after TGF-β and IGF-1 [44] was reported, and the increase in CD16 cells after transplantation should be further explored.

Artificial neural network analysis suggests that consuming more than 50% of the prescribed dose seems to be an independent protective factor for malnutrition and one-year survival after alloHSCT.

## 5. Conclusions

In this exploratory and prelminary study, an early and prophylactic oral nutritional support, started at admission to day 28 from transplantation, reduced the incidence of severe malnutrition after four weeks (PG-SGA C) in patients submitted to allo-HSCT without side effects. TE-OPF seems to have a protective role also in the prevention of gastrointestinal GVHD and infections, possibly due to anti-inflammatory, immunomodulating, and trophic effects on the gastrointestinal tract and microbiota, but due to the investigative nature of the study, larger randomized studies are warranted.

## Figures and Tables

**Figure 1 nutrients-14-03589-f001:**
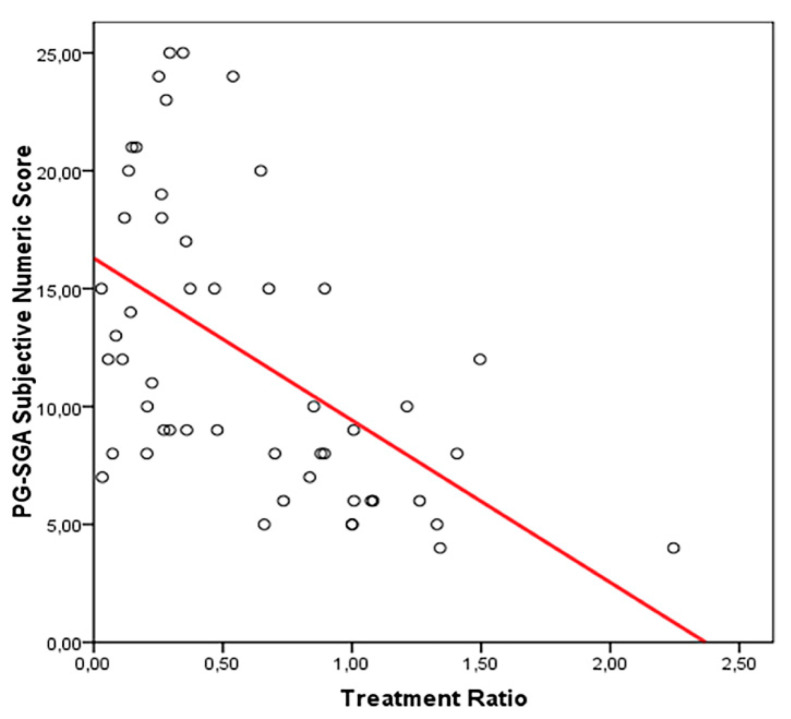
Linear correlation between PG-SGA subjective numeric score and TE-OPF intake.

**Figure 2 nutrients-14-03589-f002:**
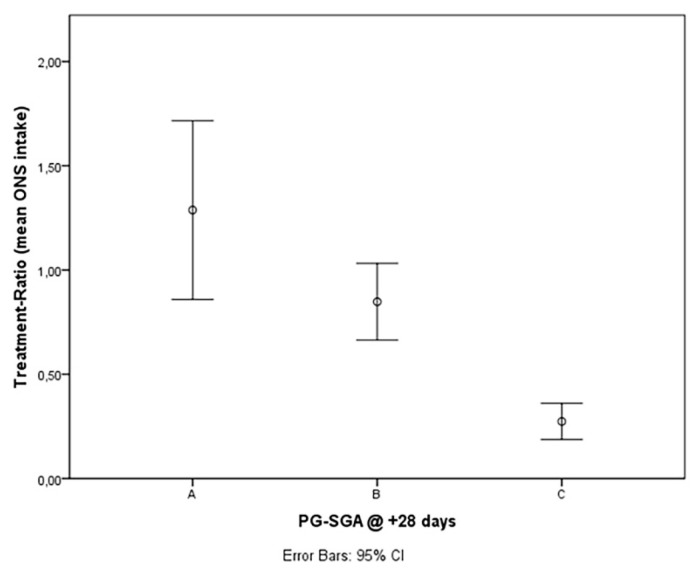
Mean TE-OPF assumption and malnutrition categories according to PG-SGA at 28 days after transplant (A = 129%, B = 85%, C = 27%, difference between A and B, *p* = 0.045, difference between A + B and C, *p* = 0.000; Mann–Whitney Test).

**Figure 3 nutrients-14-03589-f003:**
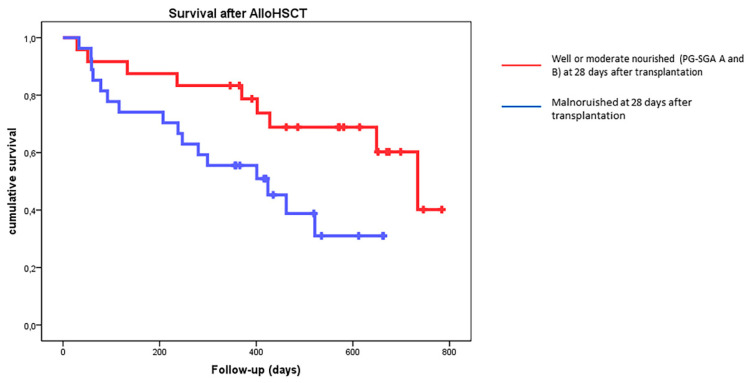
Estimated overall survival (OS) of the whole population according to PG-SGA status at 28 days after transplantation. *p* = 0.03.

**Figure 4 nutrients-14-03589-f004:**
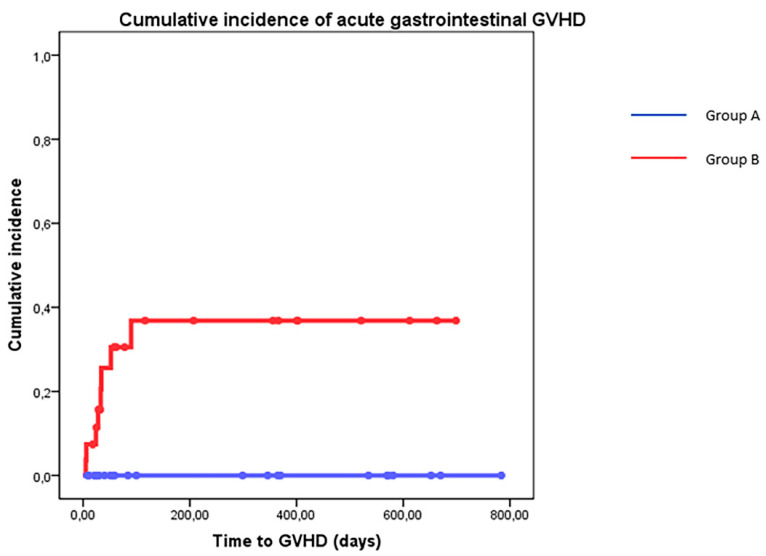
Cumulative incidence of acute GVHD according to treatment group (Group A = blue line > 50% prescribed dose of TE-OPF—Group B = red line < 50% prescribed dose of TE-OPF); *p* = 0.006.

**Figure 5 nutrients-14-03589-f005:**
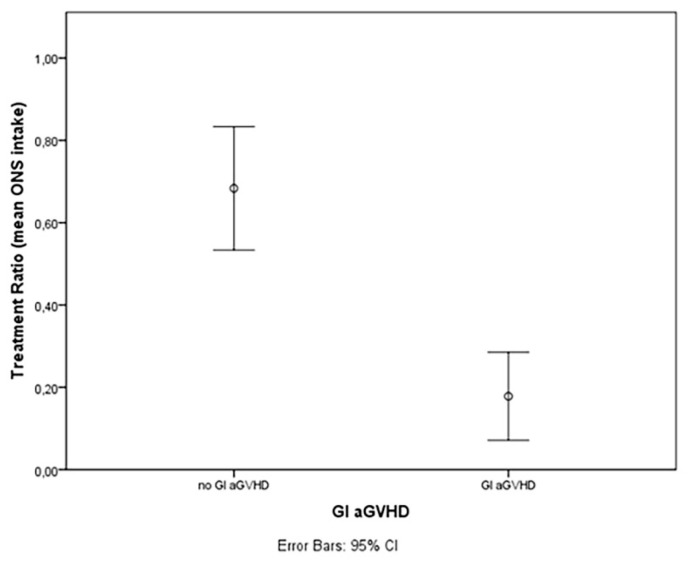
Mean TE-OPF assumption in patients with gastrointestinal GVHD compared with patients without GVHD (17.8% vs. 68.3%, *p* = 0.002, Mann–Whitney test).

**Table 1 nutrients-14-03589-t001:** Patients’ characteristics.

Sex (*n*; *Female/Male*)	22/29 (*43.1%/56.9%*)
Age (*Median*)	55 (range 20–72)
Median follow-up (*days*)	279 (29–564)
Diagnosis (*Acute leukemias, Myelodysplastic syndrome* or *Myeloproliferative disease* vs. *Lymphoma* or *Myeloma*)	40/11 (*78.5%/21.5%*)
Disease status at admission (*Complete remission/Minimal residual disease/Advanced disease*)	22/21/8 (*43.2%/41.1%/15.7%*)
Donor type (*Match-related donor/MUD/Haploidentical*)	21/20/10 (*41.2%/39.2%/15.7%*)
Stem cell source (*Peripheral blood /Bone marrow*)	48/3 (*94.1%/5.9%*)
Nutritional status at admission following PG-SGA (*Score A/B/C*)	38 A/12 B/1 C (*74.5%/23.5%/2%*)

**Table 2 nutrients-14-03589-t002:** Basal characteristics according to the Treatment Ratio (TR).

	GROUP A	Group B	*p* Value
Total (n)	24/51 (47%)	27/51 (53%)	
Sex (F/M)	9/15	13/14	ns
Age (median)	54	54	ns
Diagnosis (AL-MDS-MPD/LPD)	19/5	21/6	ns
Disease status (AD/CR)	10/14	12/15	ns
Donor (MRD/MUD/Haplo)	10/11/3	11/9/7	ns
Source (PB/BM)	22/2	26/1	ns
Conditioning (MA/RIC)	18/6	22/5	ns
PG-SGA (A/B/C)	19/5/0	19/7/1	ns

F = female, M = male, AL = Acute Leukemia, MDS = Myelodysplastic Syndrome, MPD = Myeloproliferative Disease, LPD = Lymphoproliferative Disease, AD = Advanced Disease, CR = Complete Remission, MRD = Matched Related Donor, MUD = Matched Unrelated Donor, Haplo = Haploidentical Related Donor, PB = Peripheral Blood, BM = Bone Marrow, MA = Myeloablative Conditioning, RIC = Reduced Intensity Conditioning, PG-SGA = Patient-Generated Subjective Global Assessment.

**Table 3 nutrients-14-03589-t003:** Outcomes following treatment (A > 50% TR, B < 50% TR).

	GROUP A	GROUP B	*p* Value
Total (n)	24/51 (47%)	27/51 (53%)	
PG-SGA Score A + B at +28 days (n) (%)	21/24 (87.5%)	3/27 (11.1%)	0.000
PG-SGA Score C at +28 days (n) (%)	3/24 (12.5%)	24/27 (88.9%)	0.000
Prevalence of aGVHD (%)	7/24 (29.1%)	14/27 (51.8%)	Ns
Prevalence of Gastrointestinal aGVHD (n) (%)	0/24	8/27 (29.6%)	0.005
Incidence of Sepsis (%)	7/24 (29.1%)	15/27 (55.5%)	Ns
Incidence of Pneumonia (%)	3/24 (12.5%)	13/27 (48.1%)	0.006
Survival after alloHSCT (median, days)	734 (580–881)	424 (347–501)	Ns

**Table 4 nutrients-14-03589-t004:** Levels of IGF-1 14 and 18 days after alloHSCT (ng/mL) and total lymphocytes and B, T, and NK cells (mature CD16+ and immature CD56+) 28 days after alloHSCT (cells/microliter).

Biomarker	Group A Mean (C.I.)	Group B Mean (C.I.)	
**IGF1_14**	**165.56 (45–300)**	**109.16 (36–228)**	**0.021**
**IGF1_28**	**153.64 (80.2–256)**	**109.52 (34–211)**	**0.032**
Lymphocytes_28	820 (70–2160)	490 (10–1970)	NS
**CD3+/CD4+_28**	**121.23 (3–424)**	**63.08 (1–260)**	**0.025**
CD19+_28	197.86 (0–199)	136.17 (0–418)	NS
CD56+_28	230.74 (37–1431)	138.22 (0–446)	NS
**CD16+_28**	**155.61 (30–501)**	**80.87 (0–218)**	**0.03**

## Data Availability

Raw data are available from the corresponding authors upon request.

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
