# Peer review of "Malnutrition Prevention after Allogeneic Hematopoietic Stem Cell Transplantation (alloHSCT): A Prospective Explorative Interventional Study with an Oral Polymeric Formulation Enriched with Transforming Growth Factor Beta 2 (TGF-β2)"

_nutrients, 2022, doi:10.3390/nu14173589_

Round 1

Reviewer 1 Report

Authors show an interesting and important study about malnutrition and unique way of its prevention after allo-HSCT. Patients after transplantation are special group of patients, with many risk factors of malnutrition, especially related with GvHD. Although study revelas promising effect, please concern my suggestions:

1) please explain abbreviations when used for the first time (starting from 'OS' line 28, 'GvHD' line 31, aGvHD, ONS, CSE, BMR and others later);

2) please check the significance of the sentence in lines 31-33; if patients with inadequate TE-OPF consumption had higher rish of GvHD (group B) and IGF-1 was reportd to be higher in this group, how come the incidence of aGvHD was lower? in discussion you say that IGF-1 can be protective from GvHD (lines 256-257) or you should separate aGvHD from chronic GvHD cases;

3) please correct Table 1: 'linfoma' or adevance'; please write short figure's captions in the manuscript with abbreviations explanation; this especially is related with Table 2;

4) Figure 1 caption seems to not have end;

5) can you be more precise in results analysis? in lines 170-173 please clearly state that you mean groups with PG-SGA scores A-C, not tested group A or B;

6) please correct the caption under X axis on Figure 2 (what for '@')?

7) please explain the meaning of aGvHD Magic B (line 188);

8) if something did not reach significance, please be more careful with results analysis: 'sepsis was more frequent in group B patients' (line 203-204), but was not statistically significant;

9) please use allo-HSCT word through the manuscript, not 'bone marrow transplantation' (lines 233-244);

10) why FIGURE 1 is written in capital letters? (line 247;

11) please try to extend discussion, why your study is so important and please try to add more references to data already mentioned in this part of the manuscript.

Author Response

Changes according to reviewer 1 suggestions are highlighted in red in the new text. In violet the language optimization.

Reviewer 2 Report

The findings reported herein may prove to be of value if they stimulate sufficient interest to prompt the more formal and rigorous type of investigation suggested in the concluding sentence of the submission (line 272).  At this stage, however, I would think the objective is to produce a report that will attract the attention of other investigators and, with apologies, much must be done with this submission if that goal is to be achieved.  With this in mind, please note the following points of concern.

1) This submission is a report of an uncontrolled (hence unblinded), non-randomized product-testing trial based on a small number of subjects all from a single clinical setting.  These points of design and purpose are acknowledged, either directly or indirectly, within the text of the submission but, because of these features, the findings must be regarded as preliminary.  Although this, too, is acknowledged (line 252), it is important to revise the manuscript to ensure that this point comes across clearly and emphatically.  By way of revision, the preliminary nature of the report should be stated explicitly in several places throughout the manuscript, e.g. in the abstract, in the statement of objective, in the Conclusions (Section 5) - and definitely in the title.

2) Further to item 1 in this list of concerns, a clear basis is necessary for the decision to use historical controls as this is a practice that is almost always to be discouraged.  Further, no information is provided as to the source regarding the historical controls.  This would provide, for example, the basis on which the incidence of severe malnutrition was stated to be 75% at +28 days after transplantation (lines 90-91).  A source that readers can examine for themselves is much needed.

3) No information is provided regarding the procedures used to produce the data summarized in Table 4 (IGF-1 concentrations and cell counts within blood lymphocyte subsets).

4) With apologies, the Discussion section should be deleted.  It contributes nothing insightful to the manuscript and, for the most part, it is a repetition of the objective and main findings already presented in earlier sections of the report.  The research objective was straightforward and appears to have been met within the limitations of the design.  For the purpose of a preliminary investigation, it is appropriate and sufficient simply to state this in a couple of sentences, as is done in the Conclusion (Section 5) together with the brief comment (line 272) that "larger randomized studies are warranted".  An explicit acknowledgement of the limitations of the study (because of which it is only preliminary in nature) also should be provided here.

5) lines 190-191: The percentage values cited here differ from the values presented in Table 3.

6) lines 192-193: Please cite the statistical p value (shown neither here nor in Figure 5).

7) lines 203-204: This statement is incorrect both according to Table 3 and according to the numbers shown in this sentence .  In fact, only pneumonia occurred more frequently in group B patients.  The outcome of the statistical analysis is clear, i.e. that there was no difference in the incidence of either enteritis or sepsis.

8) lines 216-217: It is confusing, at best, to make claims that the statistical analysis does not support.  The only differences identified formally were in the T cell compartment and the CD16+ NK cells (Table 4).

9) Further to items 5 through 8 in this list of concerns, it only adds tedium to the document to repeat numbers (including p values) in the text that are presented in the Tables and Figures.  It would be very helpful to readers to revise accordingly. 

10) Table 4: No measures of dispersion are presented and no units are given.  Without this information, the numbers are impossible to understand or interpret.  I note, also, that the title of the Table refers only to the analyses of cellular subsets. 

11) line 147: A citation numbered "49" is shown here, but there are only 35 items in the list of references.

12) lines 177-182: If survival analysis was performed for only the first year after transplantation, then how were the survival outcomes ranging from 456 days to 841 days determined?  Clearly some important information is missing here.

13) line 212: A clear basis is needed for the decision to assign 100 ng/mL as a critical cut-off for IGF-1 concentrations. 

14) lines 213-214: Are these outcomes of correlation analyses?  Please clarify.

15) line 221: A clear basis is needed for the decision to assign the number, 100/uL, as a critical cut-off for the CD16+ NK cell counts.

16) line 186: This subsection is identified as "3.1", but the preceding material in the Results section (lines 148-185) has no subsection designation.  From an organizational standpoint, this is confusing.

17) Readers need to know the basis for the percentages of TDEE used to define the prescribed doses of supplement (lines 134-138).  This is clearly a key point on which the interpretation of findings ultimately depends.  Likewise a clear basis is needed for the decision (lines 92-93) to set 50% of the prescribed dose of supplement as a critical cut-off.  [Why not 60% or 75% or some other percentage?]

18) There are far too many abbreviations, some of which are not defined (at least as far as I can tell).  This will discourage many readers and, hence, diminish the impact of the work.  In addition to eliminating all abbreviations except those that are needed throughout the document, it is imperative that a list of abbreviations be compiled and presented prominently in the manuscript - preferably in front of the Introduction section.

19) Much improvement is needed in the quality of language.  Please accept my apologies for drawing attention to this problem as I am sympathetic to the difficulties of attempting to communicate in a foreign language.  Nevertheless, if left unaddressed, this serious weakness in the submission will limit the impact of this piece of work.  Please note the following:

      a) It will be important to examine the manuscript thoroughly to ensure that words selected actually convey the intended meaning.  For example, the words "assumption" and "assuming" appear repeatedly when the intended meaning, I suspect, is "consumption" or "consuming".  Similarly, at least on occasion, the word "proposed" is used where, I expect, the intended meaning is "offered".  There will be other examples of this type of problem.  For example, what is meant by "major" in line 157 ("greater"?), by "conditioning" in line 163 and by "normalized importance" in lines 227 and 229?   

      b) There are a number of statements that I cannot interpret but which address points of some importance.  Examples (not an exhaustive list) include lines 96-97, lines 153-155, lines 226-230 and lines 246-249. 

20) line 278: Important and necessary details of this grant are needed, e.g. title and grant number.

21) line 314: The literature citation appears incomplete.

Author Response

Text modifications according to reviewer 2 are highlighted in blue. References were completely updated according to MDPI standards.

Reviewer 3 Report

This article reports on a prospective inventional study on treatment of alloHSCT patients with TGF-beta 2 for the prevention of malnutrition. Malnutrition remains a significant risk in alloHSCT patients and can be particularly severe in cases of acute gastrointestinal GvHD, which resembles inflammatory bowel disease (IBD). IBD is treated with TGF-b2, so the investigators tested potential effects of TGF-b2 treatment for alloHSCT patients. The authors find that TGF-beta2 treatment reduces malnutrition in alloHSCT patients, with some evidence consistent with the mechanism involving a reduction in GI aGvHD. The study is appropriately designed and reported. Overall, this report indicates that TGF-b2 treatment may have a protective role in preventing GI GvHD in a manner consistent with its expected role as an anti-inflammatory immuno-modulator. However, further study is clearly warranted. 

Author Response

English language was hopefully improved.